# Comparative Analysis of Muscle Metabolome and Amino Acid Profiles in All-Female Rainbow Trout (*Oncorhynchus mykiss*) from Different Germplasm Sources

**DOI:** 10.3390/biology14111613

**Published:** 2025-11-18

**Authors:** Tianqing Huang, Baorui Cao, Yunchao Sun, Enhui Liu, Wei Gu, Kaibo Ge, Gaochao Wang, Junran Tan, Guoqing Pan, Fuyang Bi, Datian Li, Peng Fan, Gefeng Xu

**Affiliations:** 1State Key Laboratory of Mariculture Biobreeding and Sustainable Goods, Heilongjiang River Fisheries Research Institute, Chinese Academy of Fishery Sciences, Harbin 150070, China; huangtianqing@hrfri.ac.cn (T.H.); b240501020@neau.edu.cn (B.C.); sunyunchao@hrfri.ac.cn (Y.S.); liuenhui@hrfri.ac.cn (E.L.); guwei@hrfri.ac.cn (W.G.); gekaibo@hrfri.ac.cn (K.G.); gaochaowang@ymail.com (G.W.); s230502055@neau.edu.cn (J.T.); lidatian@hrfri.ac.cn (D.L.); fanpeng@hrfri.ac.cn (P.F.); 2College of Animal Science and Technology, Northeast Agricultural University, Harbin 150030, China; 3Burqin Ertix River Endemic Cold-Water Fish Breeding and Development Co., Ltd., Altai 836699, China; yiyyiziwei1@outlook.com; 4Xinjiang Sailake Fisheries Science and Technology Development Co., Ltd., Bortala 833599, China; yyyy1231106@163.com

**Keywords:** meat quality, muscle metabolism, rainbow trout, all-female, different germplasms

## Abstract

This metabolomics study compared muscle composition in three all-female rainbow trout strains: Chinese “All-Female No. 1,” and Spanish and Danish strains. Analysis of 2198 metabolites identified 228 differential compounds, primarily organic acids, benzene derivatives, and amino acids. Enrichment revealed the phenylalanine, tyrosine, and tryptophan biosynthesis pathway as most significantly affected. Targeted quantification identified 11 key differential amino acids, with L-tyrosine, tryptamine, and L-phenylalanine being crucial for this pathway. The research provides essential data for evaluating nutritional value and supports future breeding for muscle quality in rainbow trout.

## 1. Introduction

The rainbow trout (*Oncorhynchus mykiss*) is an economically important fish species belonging to the order Salmoniformes, family Salmonidae, and genus Oncorhynchus [1]. Native to the rivers, lakes, and mountain streams along the Pacific coast of North America, it has become one of the key species promoted by the FAO for aquaculture and is widely farmed globally [2]. It has emerged as a mainstream cultivated species in countries such as the United States, Chile, Denmark, and Iran [3]. As the largest freshwater-cultivated variety in the global salmon industry, rainbow trout possesses both nutritional value and commercial significance [4]. China has over 60 years of history in rainbow trout farming, achieving a series of technological innovations—from “independent breeding of genetic resources” to “controlled breeding of triploid varieties” and further to “applications in deep-sea aquaculture scenarios”—thereby restructuring the entire industrial chain development layout. However, no study has yet systematically analyzed the muscle composition of rainbow trout farming products derived from domestic genetic sources compared to those introduced from foreign sources [5,6].

A comparative analysis of the muscle composition between rainbow trout derived from domestic germplasms and those from foreign-sourced strains holds significant multifaceted importance, which can be articulated as follows: (a) Scientific evaluation of key economic traits [7]. Through comparative analysis, it is possible to objectively assess the differences in core economic traits—such as muscle yield and nutritional quality—between domestic germplasms (e.g., selectively bred local varieties) and introduced strains (e.g., lineages imported from countries such as Spain and Denmark) [8,9]. (b) Identification of unique advantages of domestic germplasms [10]. For instance, domestic rainbow trout may exhibit distinctive characteristics in specific unsaturated fatty acids (e.g., EPA and DHA), mineral content, or flavor-related amino acids, thereby fostering a competitive niche against imported products. (c) Elucidation of genetic influence on muscle composition [11,12]. By analyzing muscle composition differences among various genetic sources under identical farming conditions, it becomes possible to determine which nutritional variations are primarily attributable to genetic background. (d) Accelerated breeding through biomarker discovery [13,14]. The analysis can help identify key metabolites or nutritional components associated with desirable traits such as high quality and yield, which may serve as molecular markers to expedite the breeding process. In summary, conducting a systematic comparison of muscle composition between domestic and foreign-sourced rainbow trout is not merely a subject of basic scientific inquiry but also a critical step toward ensuring germplasm security in China, enhancing industrial competitiveness, optimizing aquaculture practices, and guiding consumer markets. The findings from such research will directly benefit all segments of the rainbow trout industry and contribute profoundly to the high-quality and sustainable development of Chinese rainbow trout industry.

Metabolomics involves the systematic analysis of all small molecule metabolites (<1000 Da) within an organism, directly reflecting its physiological status, nutritional condition, and responses to environmental changes [15,16]. In the analysis of muscle tissues in aquatic animals, it goes beyond traditional nutrient measurements (such as crude protein and crude fat) to delve into molecular mechanisms underlying flavor formation, quality variations, stress responses, dietary effects, and species identification. For instance, a GC-MS-based metabolomic study on the taste differences between lake-cultured and pond-cultured Chinese mitten crabs (*Eriocheir sinensis*) objectively explained, at the molecular level, why consumers perceive lake-cultured crabs as having superior flavor, thereby providing a scientific basis for quality traceability and pricing [17]. Another metabolomic investigation into the effects of replacing fishmeal with plant-based protein sources in juvenile turbot (*Scophthalmus maximus*) revealed that the plant protein substitution group exhibited decreased levels of lysophosphatidylcholine (LPC), reduced content of certain essential amino acids, and markers of bile acid metabolism disruption. These findings offer precise targets for optimizing feed formulations and incorporating specific additives (such as supplementing limiting amino acids or protecting bile acid metabolism) [18]. Furthermore, metabolomic research on the impact of transport stress on fish muscle metabolism demonstrated that stress activates the “fight-or-flight” response, leading to anaerobic glycolysis of glycogen and lactate accumulation (affecting pH and tenderness), along with rapid ATP degradation generating inosine monophosphate (IMP, which increases initially and then decreases). These metabolic shifts ultimately contribute to the deterioration of muscle quality.

This study employed liquid chromatography–tandem mass spectrometry (LC-MS/MS) to conduct non-targeted metabolomic and targeted amino acid quantitative analyses on the variety “All-Female No. 1” rainbow trout, alongside all-female rainbow trout strains from Spain and Denmark, all cultivated under identical flowing pond conditions. Using chemometric techniques such as principal component analysis (PCA) and orthogonal partial least squares–discriminant analysis (OPLS-DA) for in-depth data processing and interpretation, the research aims to systematically summarize and compare the metabolic profile characteristics among sample groups of different genetic origins. KEGG enrichment analysis was subsequently performed on the filtered differentially expressed metabolites. The metabolomic analytical methodology established and the results obtained in this study will provide important data references and a scientific basis for further evaluation of the nutritional value and subsequent related research on rainbow trout from different germplasm sources.

## 2. Materials and Methods

### 2.1. Experimental Animals

#### 2.1.1. Animal Ethics

All procedures involving fish in this study were conducted in accordance with the experimental animal ethical guidelines and protocols of the Heilongjiang River Fisheries Research Institute, Chinese Academy of Fishery Sciences. The experimental protocols were reviewed and approved by the Institutional Animal Welfare and Ethics Committee of the Heilongjiang River Fisheries Research Institute (Approval No. 20250408-001).

#### 2.1.2. Animal Preparation and Sample Collection

The germplasm resources of the three species were obtained from different sources: the independently selected and bred new variety of rainbow trout “All-Female No. 1” from the Heilongjiang River Fisheries Research Institute, the all-female strain from Spain, and the all-female strain from Denmark. All fish were reared in flowing water ponds at the Bohai Cold-water Fish Experiment Station of the Heilongjiang River Fisheries Research Institute (cistern: 13 × 5 × 1.2 m). Three individuals (female, 2.3 ± 0.2 kg) were collected from each group, and muscle tissue was sampled from below the dorsal fin and adjacent areas. For each individual, 10 g of muscle tissue were collected for subsequent untargeted and targeted metabolomic analyses.

### 2.2. Non-Targeted Metabolomics Analysis

A 20 mg sample was mixed with 400 μL of an internal standard solution containing 70% methanol, homogenized, and incubated on ice for 15 min. After centrifugation at 12,000 rpm and 4 °C for 10 min, 300 μL of supernatant was collected and allowed to stand at −20 °C for 30 min. The supernatant was then centrifuged again under the same conditions, yielding a final volume of 200 μL for LC-MS analysis.

Chromatographic separation was performed using a Thermo Scientific Q Exactive HF-X mass spectrometer coupled with a vanquish liquid chromatography system (LC: Q Exactive HF-X; MS: Vanquish, Thermo Scientific, Waltham, MA, USA). Separation was achieved on a Waters ACQUITY Premier HSS T3 column (1.8 µm, 2.1 mm × 100 mm; Waters Corporation, Milford, MA, USA) maintained at 40 °C, with a flow rate of 0.4 mL/min and an injection volume of 4 μL. The mobile phase (A: 0.1% formic acid in water, B: 0.1% formic acid in acetonitrile) was eluted using the following gradient program: an increase in phase B from 5% to 20% within 2 min, followed by a rise to 60% over 3 min, then to 99% within 1 min, holding at 99% for 1.5 min, and finally returning to 5% in 0.1 min with re-equilibration for 2.4 min [19].

Mass spectrometric detection was conducted using an electrospray ionization source in both positive and negative ion modes (Vanquish, Waltham, MA, USA), with a full scan range of *m*/*z* 75–1000 and a resolution of 35,000. The ion source parameters were set as follows: ion spray voltage at 3.5 kV for positive mode and 3200 V for negative mode, sheath gas at 30 arb, auxiliary gas at 5 arb, ion transfer tube temperature at 320 °C, and vaporizer temperature at 300 °C. Data acquisition was performed using a full MS scan combined with data-dependent MS^n^ switching mode. The collision energies were set at 30, 40, and 50 V, with a signal intensity threshold of 1 × 10^6^ cps. The “Top N vs. Top Speed” was set to 10, and the dynamic exclusion time was 3 s [20].

### 2.3. Amino Acid Composition Analysis

A 500 μL aliquot of pre-cooled (−20 °C) 70% methanol solution was added to 50 mg of the sample. After thorough mixing, the mixture was centrifuged at 4 °C and 12,000 r/min for 10 min, and 300 μL of supernatant was collected. The supernatant was allowed to stand at −20 °C for 30 min, followed by another 10-min centrifugation under the same conditions. Then, 200 μL of the supernatant was passed through a protein precipitation plate for purification and subsequently analyzed using an LC-ESI-MS/MS system (LC: AB Sciex, Framingham, MA, USA; ESI: AB Sciex, Framingham, MA, USA; MS: QTRAP^®^ 6500+ System, SCIEX, Framingham, MA, USA). The system consisted of an ExionLC AD ultra-high-performance liquid chromatograph coupled with a QTRAP^®^ 6500+ mass spectrometer (SCIEX, Framingham, MA, USA) [21].

Chromatographic separation was performed on an ACQUITY BEH Amide column (1.7 μm, 100 mm × 2.1 mm internal diameter; Waters Corporation, Milford, MA, USA) maintained at 40 °C. The mobile phase consisted of an aqueous solution containing 2 mM ammonium acetate and 0.04% formic acid (phase A) and an acetonitrile solution containing 2 mM ammonium acetate and 0.04% formic acid (phase B), with a flow rate set at 0.4 mL/min and an injection volume of 2 μL. The gradient elution program was as follows: 90% phase B was maintained from 0 to 1.2 min, decreased to 60% phase B by 9 min, further decreased to 40% phase B between 10 and 11 min, rapidly returned to 90% phase B at 11.01 min, and maintained until the end of the 15-min run [22].

Mass spectrometric analysis was conducted using an electrospray ionization source with the ion source temperature set at 550 °C. The ionization voltage was 5500 V in positive ion mode and −4500 V in negative ion mode, with a curtain gas pressure of 35 psi. Detection was performed in multiple reaction monitoring mode, with declustering voltage and collision energy parameters optimized for the target compounds to ensure sensitivity and specificity [23,24].

### 2.4. Statistical Analysis

Data are expressed as mean ± standard deviation. All statistical analyses were performed using DPS statistical software (version 18.10). After initial preprocessing by unit variance scaling, unsupervised principal component analysis was conducted using the prcomp function in R language 3.5.1 (https://www.r-project.org; Accessed: 15 May 2025). Differential metabolites were screened based on a variable importance in projection (VIP) value greater than 1 and a *p*-value less than 0.05 (based on analysis of variance). VIP values were extracted from the OPLS-DA model and calculated using the R package MetaboAnalystR 1.0.1. Prior to OPLS-DA modeling, the data underwent log_2_ transformation and mean centering, and the model’s validity was verified through 200 permutation tests to avoid overfitting. Additionally, hierarchical clustering analysis was performed and Pearson correlation coefficients were calculated using the R package pheatmap. For KEGG pathway enrichment analysis, Fisher’s exact test was employed to evaluate the significance of metabolite distribution in each pathway, with all metabolites used as the background. KEGG bubble diagrams were generated using the online software MetaboAnalyst 6.0 (https://www.metaboanalyst.ca/home.xhtml; Accessed: 10 July 2025) [25].

## 3. Results

### 3.1. Classification of Metabolite Categories

In this study, muscle samples from three different sources of rainbow trout were analyzed using liquid chromatography–mass spectrometry (LC-MS). Following mass spectral peak extraction and calibration, a total of 2198 metabolites were detected. A circular diagram (Figure 1) visualizing these metabolites revealed that the top seven metabolite categories by proportion were as follows: Benzene and substituted derivatives (354 species, accounting for 16.11%); Organic acids and their derivatives (336 species, accounting for 15.29%); Amino acids and their metabolites (245 species, accounting for 11.15%); Heterocyclic compounds (224 species, accounting for 10.19%); Aldehydes, ketones, and esters (189 species, accounting for 8.60%); Fatty acids (158 species, accounting for 7.19%); and Glycerophospholipids (143 species, accounting for 6.51%).

### 3.2. Screening of Differential Metabolites

Principal Component Analysis (PCA) reduces the dimensionality of complex, high-dimensional data while retaining maximal original information, thereby establishing reliable mathematical models to describe and summarize the metabolic characteristics of the studied subjects. Figure 2A displays the PCA score plot of the three experimental muscle groups and quality control (QC) samples. The clustering of QC samples indicates excellent data stability. PC1 (*x*-axis) accounts for 15.07% of the total variance, while PC2 (*y*-axis) accounts for 7.85%. Meanwhile, Figure 2B illustrates the variance associated with PC1 and subsequent principal components. The variance values plateau after PC3, suggesting that PC1 and PC2 represent genuine biological signals, consistent with the expectations of dimensionality reduction via PCA. Overall, the results demonstrate effective separation among Groups A, B, and C, though there remains room for improvement in intra-group compactness and inter-group separability.

Although PCA efficiently extracts primary information, it is less sensitive to variables with low correlations. In contrast, Partial Least Squares–Discriminant Analysis (PLS-DA) can screen marker metabolites from extensive metabolomic datasets and establish accurate discriminant models. Orthogonal Partial Least Squares–Discriminant Analysis (OPLS-DA) was further applied to classify the three tested muscle samples, yielding cumulative values of R^2^X (cum) = 0.34, R^2^Y (cum) = 0.999, and Q^2^ (cum) = 0.575 (Appendix A). Similar to the PCA model, clear separation was observed among Groups A, B, and C. However, the OPLS-DA score plot demonstrated higher clustering density and classification clarity (Figure 2C), indicating that OPLS-DA outperforms PCA in distinguishing different samples, with high reliability and predictive capability.

Additionally, the S-plot from OPLS-DA (Figure 2D) displays the covariance between principal components and metabolites on the *x*-axis and the correlation coefficients between principal components and metabolites on the *y*-axis. Metabolites with VIP values greater than 1.0 are highlighted in red. Metabolites meeting the criteria of VIP > 1.0 and *p*-value < 0.05 (based on ANOVA) were selected, and hierarchical clustering analysis was performed on the resulting differential metabolites. Ultimately, 228 differential metabolites were identified as potential biomarkers for subsequent analysis (Appendix A).

### 3.3. Analysis of Differential Metabolites

The 228 identified biomarker metabolites were subjected to Unit Variance Scaling (UV) pretreatment, and a heatmap was generated to visualize their abundance variations (Figure 3A). Groups B (Denmark), E (All-Female No. 1), and I (Spanish) exhibited distinct clustering patterns in the heatmap. The categorical results revealed that the top differential metabolites primarily belonged to organic acids and their derivatives (37 species), benzene and substituted derivatives (24 species), and amino acids and their metabolites (21 species). Additionally, a considerable number of metabolites were detected in the following categories: glycerophospholipids (20 species), nucleotides and their metabolites (20 species), heterocyclic compounds (18 species), fatty acids (16 species), and alcohols and amines (14 species).

The KEGG database provides functional annotations and interaction networks for genes, proteins, and metabolites, enabling the exploration of metabolism-related pathways. As shown in Figure 3B, all 228 detected metabolites were enriched in 39 metabolic pathways, including purine metabolism, glycine, serine and threonine metabolism, glycerophospholipid metabolism, phenylalanine, tyrosine and tryptophan biosynthesis, glycerolipid metabolism, and phenylalanine metabolism. In the KEGG enrichment bubble plot, the *Y*-axis (−Log_10_(p)) represents the significance of pathway enrichment, with higher values indicating greater significance. The *X*-axis represents the impact of metabolites on the pathway, where larger values denote stronger influences. Accordingly, differential metabolites were most enriched in Purine metabolism and exhibited the greatest impact on Phenylalanine, tyrosine and tryptophan biosynthesis.

### 3.4. LC-MS/MS-Based Targeted Quantitative Detection of Amino Acids

Targeted quantitative analysis of amino acids enabled precise quantification of the differences in muscle amino acid composition among all-female Danish rainbow trout (B), the new variety “All-Female No. 1” rainbow trout (E), and the all-female Spanish rainbow trout strain (I). The PCA plot indicated good intra-group clustering of muscle samples from the three sources, but inter-group overlap was observed, suggesting room for improvement in separability (Figure 4A, Appendix A). The constructed OPLS-DA model demonstrated better discrimination among the different samples (Figure 4B), with optimal model performance (Appendix A). Consequently, by combining the OPLS-DA model with univariate analysis (screening based on *p*-value/FDR), 11 differentially expressed amino acids were identified (Appendix A, Table 1).

L-Threonine was the predominant component in Group B, accounting for 61.33%, followed by L-Proline (23.21%). In contrast, the main component in samples E and I was L-Proline, with proportions of 46.84% and 41.14%, respectively, while L-Threonine accounted for 25.09% and 17.70%, respectively. Notably, L-Theanine was not detected in Group B.

### 3.5. Amino Acid Enrichment Analysis

The visualized heatmap of differentially expressed amino acid metabolites revealed distinct expression patterns among the three groups (Figure 5A). These differentially expressed amino acid metabolites were most enriched in the phenylalanine, tyrosine and tryptophan biosynthesis pathway and also exhibited the greatest impact on this pathway (Figure 5B). L-Tyrosine, 5-hydroxy-tryptamine, and L-phenylalanine were identified as the key amino acid metabolites influencing the phenylalanine, tyrosine and tryptophan biosynthesis pathway (Appendix A).

Compared to the total differential amino acid content, the proportions of L-Tyrosine in groups B, E, and I were 4.31%, 8.47%, and 8.39%, respectively; 5-Hydroxy-Tryptamine accounted for 0.54%, 0.86%, and 1.26%, respectively; and L-Phenylalanine constituted 2.13%, 4.67%, and 3.40%, respectively (Table 1). The impact of differentially expressed amino acid metabolites on Phenylalanine metabolism was second only to that on Phenylalanine, tyrosine and tryptophan biosynthesis, with L-Tyrosine and L-Phenylalanine being the key metabolites in this pathway (Appendix A). In summary, L-tyrosine, 5-hydroxy-tryptamine, and L-phenylalanine were the most influential key amino acid metabolites identified in this analysis.

## 4. Discussion

Rainbow trout with different genetic backgrounds exhibit distinct advantages in core economic traits, which also impart varying nutritional qualities to their muscle tissue. A total of 2198 metabolites were detected in the rainbow trout muscle samples. Benzene and substituted derivatives were the most abundant category, accounting for 16.11% of all metabolites. Certain benzene derivatives, such as benzaldehyde and phenylethanal, can enhance the flavor of fish meat [26]. Organic acids and their derivatives (336 species, 15.29%) are central to muscle movement and energy metabolism [27]. Fish muscle protein contains all essential amino acids required by the human body, offering high nutritional value [28]. Amino acids are crucial organic compounds that play vital roles in many biological processes, including protein synthesis, cell growth and development, and energy production [29]. The eating quality and nutritional value of muscle tissue largely depend on its amino acid profile and the composition of related metabolites [30,31]. The 245 detected amino acids and their metabolites accounted for 11.15% of all metabolites. The subsequent PCA, while showing some separation between groups, revealed limitations of the model due to partial overlap, prompting the introduction of the OPLS-DA model to screen for biomarker metabolites [25,32].

Purine metabolism plays a crucial role in energy supply, participating in the synthesis, degradation, and recycling of nucleic acids. Furthermore, the purine metabolism process is responsible for producing inosinate and inosine monophosphate (IMP), which are key contributors to the umami taste of fish meat. Therefore, the significant enrichment of differential metabolites in purine metabolism is directly related to the flavor experienced when consuming fish.

In the phenylalanine, tyrosine and tryptophan biosynthesis pathway, phenylalanine, tyrosine, and tryptamine—all aromatic amino acids—can be converted into volatile metabolites, enhancing flavor [33,34]. In animals, aromatic amino acids are considered “essential [35]. Phenylalanine is a precursor of tyrosine and has been shown to be associated with sugar and lipid metabolism [36,37,38]. Additionally, phenylalanine serves as the “source” of the aromatic flavor spectrum and is an essential nutrient; its thermal degradation produces aromatic compounds that enrich the flavor profile of fish meat [39]. L-Tyrosine is commonly used as a dietary supplement, primarily due to reports of its ability to stimulate brain activity for improved memory and mental alertness, act as an appetite suppressant (aiding in controlling depression and anxiety), and enhance physical performance [35]. The new variety “All-Female No. 1” rainbow trout, containing higher levels of phenylalanine and L-tyrosine, thus offers greater nutritional value.

Certain amino acids, known as umami or flavor amino acids, contribute to the sensory properties of meat by enhancing its taste profile. For example, proline, threonine, serine, glycine, and alanine can directly increase the sweetness of fish meat [40,41]. Among the 11 differential amino acids detected in this study, the combined proportions of the umami amino acids L-threonine and L-proline in the three groups reached 84.54%, 71.93%, and 58.84%, respectively. The all-female Danish rainbow trout had the highest proportion, followed by the “All-Female No. 1” variety. This indicates that the all-female Danish rainbow trout and the “All-Female No. 1” rainbow trout possess more desirable flavor characteristics.

## 5. Conclusions

This study evaluated the nutritional components in the muscle tissue of rainbow trout with three distinct genetic backgrounds, identifying a total of 2198 metabolites and 228 differential metabolites. The results revealed that the differences in nutritional composition among the rainbow trout muscle tissues from different sources were closely associated with amino acid metabolism. The all-female Danish rainbow trout and the “All-Female No. 1” variety exhibited more desirable flavor characteristics due to their high content of the umami amino acids L-threonine and L-proline. Meanwhile, the “All-Female No. 1” rainbow trout demonstrated higher nutritional value owing to its elevated levels of phenylalanine and L-tyrosine. In conclusion, genetic background directly contributes to differences in nutritional value among rainbow trout from various germplasm sources. Future studies should build on this work and explore these findings further with a larger sample size. This study provides a foundation for the future selection and breeding of superior rainbow trout strains and the development of muscle quality evaluation criteria.

## Figures and Tables

**Figure 1 biology-14-01613-f001:**
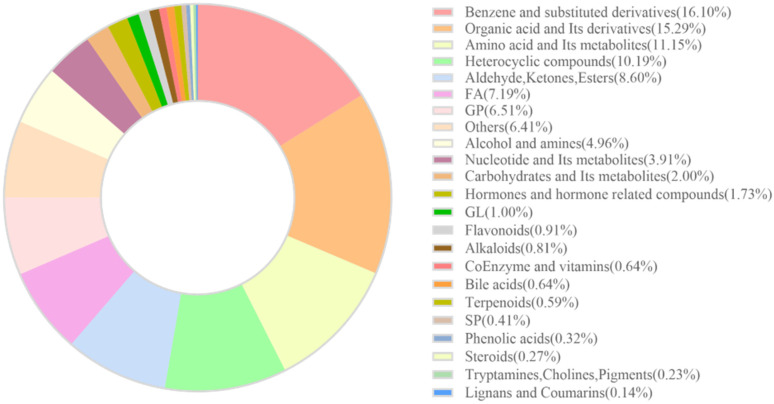
Doughnut chart illustrating the composition of metabolite categories.

**Figure 2 biology-14-01613-f002:**
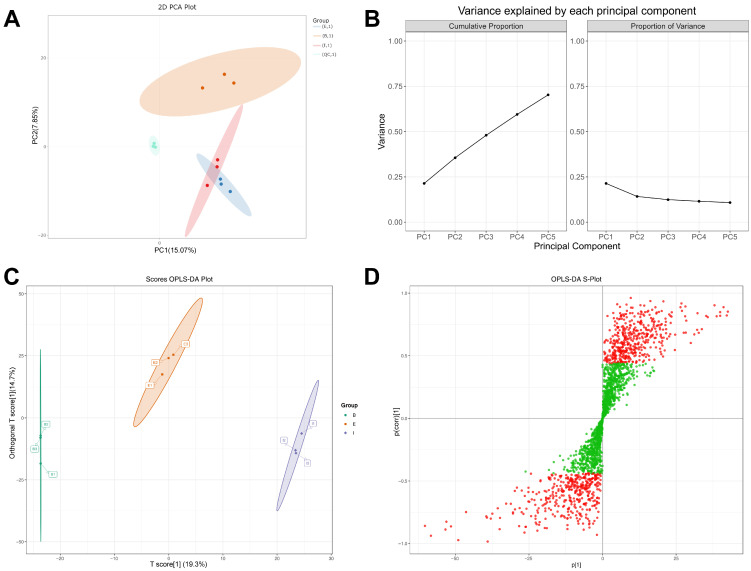
Screening of differential metabolites in all-female strain from Denmark (Group B), the “All-Female No. 1” variety (Group E), and the all-female Spanish (Group I) rainbow trout strain. (**A**) PCA Score Plot; (**B**) Component Loading Plots; (**C**) OPLS-DA Scores Plot; (**D**) OPLS-DA S-Plot.

**Figure 3 biology-14-01613-f003:**
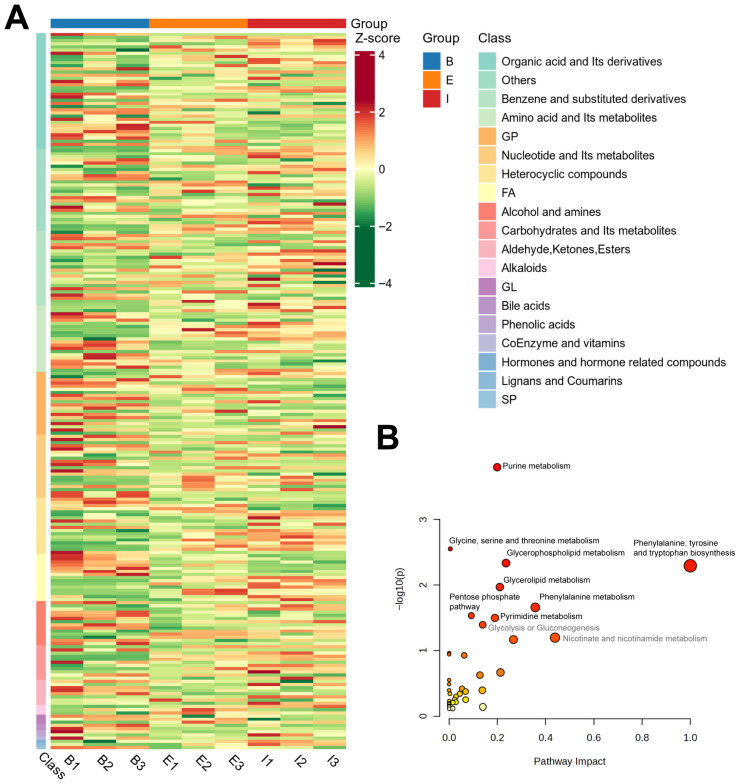
Hierarchical clustering analysis of differential metabolites (**A**) and KEGG enrichment bubble plot (**B**) of metabolites in all-female Danish (Group B), the new variety “All-Female No. 1” (Group E), and the all-female Spanish (Group I) rainbow trout strain.

**Figure 4 biology-14-01613-f004:**
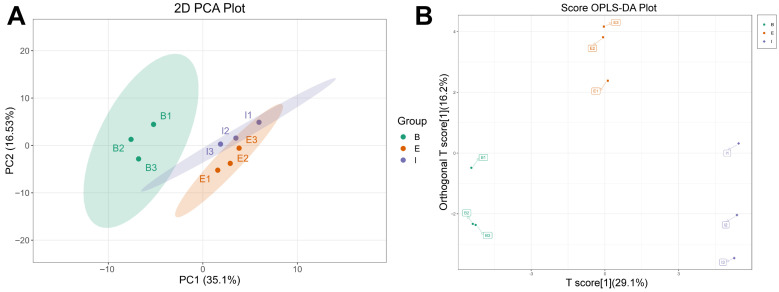
PCA score plot (**A**) and OPLS-DA scores plot (**B**) of all-female Danish (Group B), the new variety “All-Female No. 1” (Group E), and the all-female Spanish (Group I) rainbow trout strain.

**Figure 5 biology-14-01613-f005:**
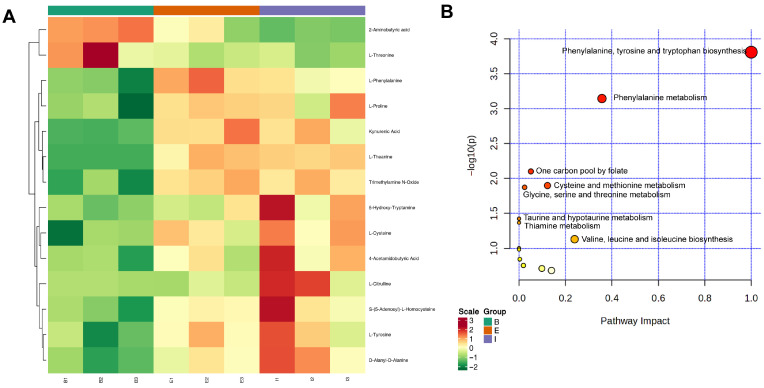
Hierarchical clustering analysis of differential metabolites (**A**) and KEGG enrichment bubble plot of amino acid metabolites (**B**) in all-female Danish (Group B), the new variety “All-Female No. 1” (Group E), and the all-female Spanish (Group I) rainbow trout strain.

**Table 1 biology-14-01613-t001:** Differential expression of free amino acid and related metabolites in the muscle of all-female Danish (Group B), the new variety “All-Female No. 1” (Group E), and the all-female Spanish (Group I) rainbow trout strain.

	B		E		I	
Compound	Amount (µg/g) ^(a)^	Proportion ^(b)^	Amount (µg/g) ^(a)^	Proportion ^(b)^	Amount (µg/g) ^(a)^	Proportion ^(b)^
L-Tyrosine	13,704.2 ± 5896.15 ^b^	4.31	28,283.23 ± 4213.57 ^a^	8.47	30,555.67 ± 8867.18 ^a^	8.39
Trimethylamine N-Oxide	20,380.28 ± 2764.38 ^b^	6.41	31,471.33 ± 1400.23 ^a^	9.42	30,737.62 ± 1823.98 ^a^	8.44
L-Threonine	195,059.61 ± 86,089.56 ^a^	61.33	83,783.74 ± 17,297.38 ^b^	25.09	64,448.99 ± 24,329.51 ^b^	17.70
S-(5-Adenosyl)-L-Homocysteine	1118.33 ± 738.46 ^b^	0.35	3113.02 ± 181.53 ^ab^	0.93	4164.81 ± 1730.88 ^a^	1.14
L-Theanine	- ^(c)^	- ^(c)^	96.88 ± 17.54 ^a^	0.03	97.06 ± 2.94 ^a^	0.03
L-Cysteine	209.9 ± 181.97 ^b^	0.07	594.08 ± 154.66 ^a^	0.18	737.82 ± 167.21 ^a^	0.20
5-Hydroxy-Tryptamine	1707.26 ± 325.97 ^a^	0.54	2887.39 ± 795.67 ^ab^	0.86	4580.22 ± 1580.99 ^a^	1.26
L-Phenylalanine	6762.19 ± 2082.21 ^b^	2.13	15,603.82 ± 1866.15 ^a^	4.67	12,387.79 ± 1061.41 ^a^	3.40
D-Alanyl-D-Alanine	60.15 ± 8.43 ^b^	0.02	90.83 ± 7.02 ^a^	0.03	109.82 ± 17.46 ^a^	0.03
L-Proline	73,828.46 ± 34,296.86 ^b^	23.21	156,420.11 ± 5158.48 ^a^	46.84	149,854.57 ± 38,342.37 ^a^	41.14
L-Citrulline	5201.7 ± 1716.9 ^b^	1.64	11,619.81 ± 8676.75 ^b^	3.48	66,542.33 ± 40,567.9 ^a^	18.27
Total	318,032.09		333,964.23		364,216.69	

^(a)^ Average ± standard deviation (µg/g); ^(b)^ Average value proportion (%); ^(c)^ Below detection limit; ^ab^ Different letters indicate significant differences among the three groups (*p* < 0.05); All the treatments are performed in triplicate.

## Data Availability

The data supporting the reported results of this study can be provided from the corresponding author upon reasonable request.

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
