# Peer review of "Comparative Analysis of Muscle Metabolome and Amino Acid Profiles in All-Female Rainbow Trout (Oncorhynchus mykiss) from Different Germplasm Sources"

_biology, 2025, doi:10.3390/biology14111613_

Round 1
Reviewer 1 Report
Comments and Suggestions for Authors
This paper, titled "Comparative Analysis of Muscle Metabolome and Amino Acid Profiles in All-Female Rainbow Trout from Different Germplasm Sources," is relatively complete in terms of title, structure, content, and data analysis, demonstrating certain potential for publication. However, the following issues need to be addressed before formal submission:
1. The study employs untargeted metabolomics and targeted amino acid metabolomics to analyze muscle composition in rainbow trout from different germplasm sources. However, it is well-known that salmonid fish are renowned for their rich fatty acid content. Therefore, the paper should incorporate relevant fatty acid analysis and discussion.
2. In the Methods section, were the three strains of fish reared under the same conditions and at the same time?
3. In Figure 2C, Group E ("All-Female No. 1") and Group I (Spanish) show a trend of overlapping, while Group B (Danish) appears more separated. This observation should be mentioned in the text, and possible reasons should be discussed.
4. Strictly speaking, compounds such as "Trimethylamine N-Oxide" and "S-(5-Adenosyl)-L-Homocysteine" are not amino acids. It is recommended to revise the title of Table 1 to "Differential Expression of Free Amino Acids and Related Metabolites" or a similar expression.
5. Note (c) in Table 1: "There is no such amino acid component" – It would be more scientifically accurate to replace this with "Below detection limit."
6. The text flows smoothly overall, though there remain a few instances where grammar and word choice could be refined.
Reviewer 2 Report
Comments and Suggestions for Authors
Dear Authors,
This manuscript, "Comparative Analysis of Muscle Metabolome and Amino Acid Profiles in All-Female Rainbow Trout from Different Germplasm Sources", has been well written.
The research paper presents a comparative metabolomic analysis of muscle tissue from three all-female rainbow trout strains: a domestically bred Chinese variety ("All-Female No. 1"), a Spanish strain, and a Danish strain, and concludes that genetic background directly contributes to differences in nutritional value among rainbow trout from various germplasm sources. However, to improve the manuscript, the following are some inputs:
TITLE
The title should describe the manuscript's content, and abbreviations should be avoided, except those commonly used; please recheck and improve.
ABSTRACT
An abstract should contain brief information on background, methods, results, and conclusion. Please recheck.
Please recheck, e.g., there is a scientific name that has not been italicized.
INTRODUCTION
This section normally presents the problem studied or the hypothesis tested, reviewing specific findings of previous related studies that are challenging, the main features of the methods used, and a brief overview of the main results of the research; please recheck.
MATERIAL AND METHODS
The methods section should be written systematically and in detail, explaining how the research was conducted. If any new procedure should be described in sufficient detail, please recheck.
The study utilized only three individuals from each of the three trout groups for sample collection. A sample size of n=3 is very small for metabolomics studies, which are known for high biological variability. This limited sample size may not adequately represent the entire population of each trout strain, potentially affecting the statistical power and the generalizability of the findings. Differences observed could be due to individual variation rather than consistent genetic differences between the strains. Please add an explanation for this limitation and this issue in the discussion section.
Please include the related references for each section of the laboratory sample analysis method.
RESULTS
Please improve the readability of Figures 2 and 4 by, for example, increasing the font size of the variables displayed on the x- and y-axes.
Please recheck and, if necessary, improve Figure 5A, as readers may find it challenging to differentiate the image colors among the scale, group, and class.
DISCUSSION
Please recheck and discuss that Principal Component Analysis (PCA) showed some limitations, although there was separation among the groups. This suggests that the metabolic differences, while present, might not be as distinct as implied by the subsequent OPLS-DA model. Similarly, for the targeted amino acid analysis, the PCA plot showed "inter-group overlap," again indicating that the separation between groups was not clear-cut.
CONCLUSION
In the conclusion, it suggests reformulating and synthesizing the primary findings and their implications, ensuring not to repeat sentences from the methods and research results sections. Briefly address the limitations and propose directions for future research. The conclusion should effectively address the research problems and objectives, as well as their consequences, making it engaging for readers.
REFERENCES
The references should contribute meaningfully to the manuscript content; please check again.
Reviewer 3 Report
Comments and Suggestions for Authors
The manuscript “Comparative Analysis of Muscle Metabolome and Amino Acid Profiles in All-Female Rainbow Trout from Different Germplasm Sources” describes a comparative metabolomic analysis of three rainbow trout strains. Generally, studies on muscle metabolomics in fish that provide useful insights into metabolic processes with potential implications for aquaculture and physiology are beneficial for industry and related research sectors. Further, the methodical and statistical approaches are well-chosen and allow for good insights into the matter. However, several aspects of the manuscript require revision before it can be considered for final publication.
Next to some minor errors in the text, the study presently misses information on the husbandry systems and parameters used for the specimen, as these influence metabolism and allow for a later comparison of the study. Furthermore, the sample tissue is not described sufficiently, as e.g., no differentiation between red and white muscle tissue portions is mentioned, even though these will create differences in the metabolomic analyses. The abstract should include the scientific relevance of the study, and throughout the text, it has to be ensured that all species names are italicised. The introduction currently focuses too heavily on the Chinese aquaculture context, which limits its appeal to an international readership. It should be broadened to include a more global perspective and supported with missing references, especially in Lines 59–65. In the results, repetitive methodological descriptions should be removed. Several figures require higher resolution and/or larger lettering to ensure readability.
As some authors are assigned to industry affiliations, the Conflict of Interest section must transparently address affiliations with commercial partners and explain how potential conflicts were managed.
Line-Specific Comments
L59–65: Add references to the statements
L127–131: Has some other external institution or legislation been included in the decision process?
L208–209, L233–236: Remove method repetitions
Table 1 (L324–326): Move legend above table.
Round 2
Reviewer 2 Report
Comments and Suggestions for Authors
Dear Author,
Thank you for your response and for the improvements to the manuscript accordingly. Additionally, please recheck and include the relevant references for each section of the laboratory sample analysis method, particularly sections 2.2 and 2.3.
Author Response
Comments 1: Additionally, please recheck and include the relevant references for each section of the laboratory sample analysis method, particularly sections 2.2 and 2.3.
Response 1: We are grateful for this important comment. We apologize for the oversight of not including the references in the Methods section in the previous version. In the revised manuscript, we have now inserted references [19] – [25] accordingly, with the changes highlighted in yellow for your convenience.